# BMU-MoCo: Bidirectional Momentum Update for Continual Video-Language Modeling

**Yizhao Gao**[1,2] **Nanyi Fei**[1,2] **Haoyu Lu**[1,2] **Zhiwu Lu**[1,2,*] **Hao Jiang**[3] **Yijie Li**[3] **Zhao Cao**[3]

[1]Gaoling School of Artificial Intelligence, Renmin University of China, Beijing, China
[2]Beijing Key Laboratory of Big Data Management and Analysis Methods
[3]Huawei Poisson Lab, Hangzhou, Zhejiang, China
{gaoyizhao, luzhiwu}@ruc.edu.cn

## Abstract

Video-language models suffer from forgetting old/learned knowledge when trained with streaming data. In this work, we thus propose a continual video-language modeling (CVLM) setting, where models are supposed to be sequentially trained on five widely-used video-text datasets with different data distributions. Although most of existing continual learning methods have achieved great success by exploiting extra information (*e.g.*, memory data of past tasks) or dynamically extended networks, they cause enormous resource consumption when transferred to our CVLM setting. To overcome the challenges (*i.e.*, catastrophic forgetting and heavy resource consumption) in CVLM, we propose a novel cross-modal MoCo-based model with bidirectional momentum update (BMU), termed BMU-MoCo. Concretely, our BMU-MoCo has two core designs: (1) Different from the conventional MoCo, we apply the momentum update to not only momentum encoders but also encoders (*i.e.*, bidirectional) at each training step, which enables the model to review the learned knowledge retained in the momentum encoders. (2) To further enhance our BMU-MoCo by utilizing earlier knowledge, we additionally maintain a pair of global momentum encoders (only initialized at the very beginning) with the same BMU strategy. Extensive results show that our BMU-MoCo remarkably outperforms recent competitors w.r.t. video-text retrieval performance and forgetting rate, even without using any extra data or dynamic networks.

## 1 Introduction

Existing video-language modeling (VLM) methods have achieved promising performance for video-text retrieval [59, 39, 61, 30, 24, 53, 28, 4] with non-streaming data. However, in real-world application scenarios, VLM models need to evolve with streaming data (*e.g.*, collected from the Internet [39, 42]) to accommodate more tasks. Under this setting, since it costs too much resource to retrain the model with both old and new data for each task, a common practice is to fine-tune VLM models with only the newly-arrived data. Note that such model fine-tuning leads to severe performance degradation on previous tasks. This is a well-documented phenomenon called catastrophic forgetting [18, 38] under the conventional continual learning setting [48, 45, 34, 16, 23, 60].

Therefore, in this work, we propose a continual video-language modeling (CVLM) setting to better simulate the realistic scenario. Under our CVLM setting, models are supposed to be sequentially trained on five widely-used video-text datasets: VATEX [54], ActivityNet [25], MSR-VTT [55], DiDeMo [20], and MSVD [10]. An evaluation protocol is also established for CVLM, which contains three metrics to respectively measure the text-to-video retrieval performance (Recall@1, shortened as

---

*The corresponding author.

36th Conference on Neural Information Processing Systems (NeurIPS 2022).

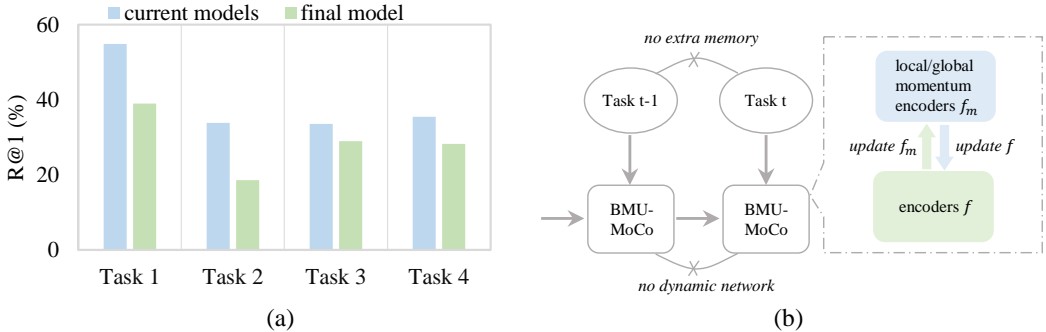

(a)

(b)

Figure 1: Illustration of the catastrophic forgetting problem in CVLM and the core design for our BMU-MoCo. **(a) The catastrophic forgetting problem in CVLM.** We train a basic cross-modal MoCo model on five tasks and present the comparative results of the final model and current models on learned tasks (Task 1–4). Note that there is no catastrophic forgetting on Task 5 and thus this task is omitted here. **(b) The core design of BMU-MoCo.** Different from the conventional MoCo, we update not only momentum encoders but also encoders through the bidirectional momentum update (BMU) strategy without extra memory or dynamic network across all tasks.

R@1), forgetting rate (FR), and harmonic mean (HM) performance (see more details of the evaluation protocol in Sec. 4.1). Moreover, we implement a basic cross-modal MoCo [19] model (Base-MoCo) as our baseline method since it has shown superior performance on video-language modeling [33, 35]. As illustrated in Figure 1(a), we can observe that in spite of achieving great R@1 results with current models (evaluated right after trained on each task), the performance of the final Base-MoCo model (trained across all five tasks) drops significantly (i.e., catastrophic forgetting).

To tackle the catastrophic forgetting problem, most recent continual learning works attempt to preserve the learned knowledge from a variety of perspectives: (1) Maintaining a memory buffer to save and exploit data from previous tasks [46, 5, 34, 3, 8, 45]; (2) Generating pseudo data of learned tasks [51, 27, 43, 57]; (3) Extending the network architecture dynamically as each new task arrives [48, 16, 1, 7, 31]. However, when these methods are transferred to our CVLM setting, the resource consumption is enlarged rapidly as the number of tasks grows, due to the characteristic of video data. In addition, another branch of continual learning works focus on imposing a regularization constraint with quadratic penalty [23, 60, 14, 49] or knowledge distillation [32, 2, 21, 44, 22], which leads to an unwanted trade-off on the performance of old and new tasks with limited neural resources [40]. Therefore, it is a long-standing and arduous challenge to train a video-language modeling network under the CVLM setting with both effectiveness and efficiency taken into consideration.

To overcome this challenge under the CVLM setting, we devise BMU-MoCo, a cross-modal MoCo-based model with a novel bidirectional momentum update (BMU) strategy. As shown in Figure 1(b), our BMU-MoCo needs neither extra memory data nor dynamically extended neural networks, which is quite different from previous continual learning methods. Concretely, similar to cross-modal MoCo applied in [33, 35], our proposed BMU-MoCo has a video encoder (*i.e.*, ViT-Base [13]) and a text encoder (*i.e.*, BERT-Base [12]), followed by the momentum video/text encoders. Different from the original MoCo [19] and its cross-modal versions [15, 33, 35] that utilize momentum update for only momentum encoders to maintain a large consistent queue, our BMU strategy imposes momentum update on both momentum encoders and (video/text) encoders. As a result, at each training step, the encoders of our BMU-MoCo learn the new knowledge by end-to-end update with back-propagation whilst reviewing the old knowledge directly from the parameters of momentum encoders by momentum update. In our opinion, our BMU-MoCo outperforms existing methods for two main reasons: (1) Momentum encoders are initialized by current encoders at the beginning of each new task and then progress slowly, which helps our model preserve adequate old knowledge without sacrificing the performance on new tasks; (2) Since there is no category information under the CVLM setting, learning from memory data or distilling with a batch of new data only absorbs *part of* previous knowledge while our BMU-MoCo learns *holistic* old knowledge directly from the parameters of momentum encoders. To further enhance our proposed BMU-MoCo, we also maintain a pair of global (cross-task) momentum encoders with the same BMU strategy, which are only initialized at the very beginning of the whole training process and thus preserve earlier knowledge than the normal local (task-specific) momentum encoders.

The main contributions of this paper are four-fold: (1) We propose a new continual video-language modeling (CVLM) setting, where models are supposed to be sequentially trained on five widely-used video-text datasets. (2) To effectively and efficiently overcome the catastrophic forgetting problem under the CVLM setting, we devise BMU-MoCo, a cross-modal MoCo-based model with a novel bidirectional momentum update (BMU) strategy. For the first time, our BMU strategy can review holistic old knowledge directly from the parameters of momentum encoders while learning on new tasks, without extra memory or dynamic network across all tasks. (3) To bring further improvements to our proposed BMU-MoCo, a pair of global momentum encoders are maintained by the same BMU strategy to preserve and review earlier knowledge under the CVLM setting. (4) Extensive experimental results demonstrate that our BMU-MoCo outperforms recent continual learning methods by large margins w.r.t. both text-to-video retrieval performance and forgetting rate, even without any extra memory data or dynamically extended networks.

## 2 Related Work

**Video-Language Modeling.** Most existing methods for video-language modeling follow two paradigms: (1) Single-stream methods [36, 61, 52, 29, 28, 56] typically include a multi-modal transformer to achieve fine-grained cross-modal interaction between the video and language modalities. Although achieving great performance, they suffer from the huge time complexity caused by the pairwise inputs during inference, which makes them unsuitable for practical applications. (2) Two-stream methods [17, 41, 4, 33, 35] learn video and text representations independently, and align them after encoding. To ensure the inference efficiency, both the baseline method (*i.e.*, Base-MoCo) and our BMU-MoCo for CVLM are set to be two-stream methods. Importantly, different from Base-MoCo, our BMU-MoCo has a novel BMU strategy to address the catastrophic forgetting problem and two extra global momentum encoders to further boost the model performance.

**Continual Learning.** Conventional continual learning methods mainly focus on image classification tasks. They can be roughly categorized into three groups: (1) *Rehearsal-based* methods apply extra memory to store sampled data [45, 34, 46, 9, 5, 3, 8, 6] or generate pseudo data [51, 27, 43, 57] from previous tasks. The memory size and the training complexity tend to be enlarged significantly as the number of tasks grows. (2) *Expansion-based* methods either add extra extended networks for new tasks [48, 16, 1, 7, 31] or select partial model parameters to update for different tasks [58, 47, 50]. They need more computational resources especially for a long sequence of training tasks (e.g., under our CVLM setting). (3) *Regularization-based* methods modify the model parameters with quadratic loss penalty [23, 60, 14, 49] or knowledge distillation constraints [32, 2, 21, 44, 22]. Although succeeded in image classification tasks, they still face a large challenge in balancing the model performance between old and new tasks when applied to our CVLM setting. Although our BMU-MoCo can be classified as a regularization-based method, it has a vital difference from existing regularization-based methods: benefiting from the bidirectional momentum updating process, our BMU-MoCo can directly utilize the holistic previous knowledge from the parameters of momentum encoders for model training (*i.e.*, updating the encoders), and *simultaneously* update the momentum encoders at each training step to accommodate new tasks.

## 3 Methodology

### 3.1 Preliminary

We propose a new continual video-language modeling (CVLM) setting, where models are supposed to be sequentially trained on $n$ video-text datasets $\mathcal{D} = [\mathcal{D}_1, \mathcal{D}_2, \cdots, \mathcal{D}_n]$. For each task $t$, it contains a dataset $\mathcal{D}_t = \{V_i, T_i\}_{i=0}^{N_t - 1}$ with $N_t$ video-text pairs, where $V_i$ denotes a video with $S_i$ frames and $T_i$ represents an English text description. The target of CVLM is to learn a video encoder $f_{\theta_V}$ and a text encoder $f_{\theta_T}$, which can respectively project the input video and its related text description into a joint embedding space with nearest metric distance. Different from the classical VLM setting which only considers the model performance on the current dataset $\mathcal{D}_t$, our CVLM setting requires the models to prevent the catastrophic forgetting on the previously-used datasets $[\mathcal{D}_1, \mathcal{D}_2, \cdots, \mathcal{D}_{t-1}]$ $(t > 1)$ while also performing well on the current dataset $\mathcal{D}_t$. Note that our proposed BMU-MoCo for CVLM is a memory-free method which only utilizes the current dataset $\mathcal{D}_t$ for each task $t$ (without the need of reloading the other datasets). Therefore, without particular statement, we only consider task $t$ with $\mathcal{D}_t$ in the following subsections for formulation simplicity.

## 3.2 Network Architecture

**Video Encoder.** We follow the most recent video-language modeling works [28, 56, 33, 35] to learn video representation by fusing the image embeddings of sampled frames per video. Concretely, given each video $V_i$ with $S_i$ frames, we randomly sample $s$ frames ($s < S_i$) and embed them with an image encoder $f_{img}$ (*i.e.*, ViT-Base [13]) to obtain the frame embeddings:

$$F_{img}^{i,j} = f_{img}(x_i^j), j = 1, 2, \cdots, s, \tag{1}$$

where $x_i^j$ denotes the $j$-th sampled frames of video $V_i$ and $F_{img}^{i,j}$ denotes its image embedding encoded by $f_{img}$. Then we project $F_{img}^{i,j}$ by a Linear layer $f_{proj}$:

$$F_{proj}^{i,j} = f_{proj}(F_{img}^{i,j}), j = 1, 2, \cdots, s, \tag{2}$$

where $F_{proj}^{i,j} \in \mathbb{R}^d$ denotes the projected $d$-dimensional image embedding of $F_{img}^{i,j}$. Following COTS [35] and HiT [33], we obtain the final video embedding of $V_i$ by adopting a fusing layer $f_{avg}$ to aggregate the image embeddings $\{F_{img}^{i,j}\}$:

$$F_V^i = f_{avg}(F_{proj}^{i,1}, F_{proj}^{i,2}, \cdots, F_{proj}^{i,j}), \tag{3}$$

where $f_{avg}$ denotes an Average Pooling layer and $F_V^i \in \mathbb{R}^d$ is the video embedding of $V_i$. In summary, our video encoder $f_{\theta_V}$ encodes the video inputs by adopting $f_{img}$, $f_{proj}$ and $f_{avg}$ in Eqs. (1)–(3).

**Text Encoder.** For the language modality, we adopt BERT-Base [12] as our backbone to encode each input text $T_i$. In detail, we first tokenize $T_i$ into a sequence of text tokens $[l_i^1, l_i^2, \cdots, l_i^{r_i}]$, where $r_i$ denotes the length of $T_i$. Then we obtain the text token embeddings through the backbone $f_{bert}$:

$$F_{bert}^i = f_{bert}(l_i^1, l_i^2, \cdots, l_i^{r_i}), \tag{4}$$

where $F_{bert}^i$ denotes the token embeddings of $T_i$ obtained by $f_{bert}$. We then project them by a Linear layer $\hat{f}_{proj}$ into the $d$-dimensional space as:

$$\begin{aligned} \hat{F}_{proj}^i &= [\hat{F}_{proj}^i[1], \hat{F}_{proj}^i[2], ..., \hat{F}_{proj}^i[r_i]] \\ &= \hat{f}_{proj}(F_{bert}^i[1], F_{bert}^i[2], \cdots, F_{bert}^i[r_i]), \end{aligned} \tag{5}$$

where $F_{bert}^i[j]$ denotes the $j$-th element of $F_{bert}^i$, and $\hat{F}_{proj}^i[j] \in \mathbb{R}^d$ represents the projected text embedding of token $j$ in $T_i$ (which has the same dimension $d$ as video embedding $F_V^i$). To obtain the final text embedding of $T_i$, we apply an Average Pooling layer $f_{avg}$:

$$F_T^i = f_{avg}(\hat{F}_{proj}^i[1], \hat{F}_{proj}^i[2], \cdots, \hat{F}_{proj}^i[r_i]), \tag{6}$$

where $F_T^i \in \mathbb{R}^d$ denotes the text embedding of $T_i$. In summary, our text encoder $f_{\theta_T}$ encodes the text inputs by adopting $f_{bert}$, $\hat{f}_{proj}$, and $f_{avg}$ in Eqs. (4)-(6).

## 3.3 BMU-MoCo

**Cross-Modal MoCo.** Similar to the original single-modal MoCo [19], recent state-of-the-art video-language modeling works COTS [35] and HiT [33] construct a cross-modal MoCo architecture to maintain video/text momentum encoders by the same momentum update mechanism, which creates consistent queues for cross-modal contrastive learning objectives. As illustrated in Figure 2, our BMU-MoCo follows this paradigm and further transfers it to our CVLM setting. Concretely, for a mini-batch of $N_B$ video-text pairs $\mathcal{B} = \{V_i, T_i\}_{i=1}^{N_B}$, we first obtain the query embeddings $q_i^V, q_i^T$ of $V_i, T_i$ by video encoder $f_{\theta_V}$ and text encoder $f_{\theta_T}$:

$$q_i^V = f_{\theta_V}(V_i), \quad q_i^T = f_{\theta_T}(T_i). \tag{7}$$

Then we maintain two momentum encoders $f_{\theta_{V,m}}, f_{\theta_{T,m}}$ (termed local momentum video/text encoders in Figure 2) for both video and text modalities, whose parameters $\theta_{V,m}, \theta_{T,m}$ are initialized by $\theta_V, \theta_T$ at the beginning of each task $t$. During the training process, $\theta_{V,m}, \theta_{T,m}$ are continuously updated by $\theta_V, \theta_T$ with the momentum update strategy:

$$\theta_{V,m} = m \cdot \theta_{V,m} + (1-m) \cdot \theta_V, \quad \theta_{T,m} = m \cdot \theta_{T,m} + (1-m) \cdot \theta_T, \tag{8}$$

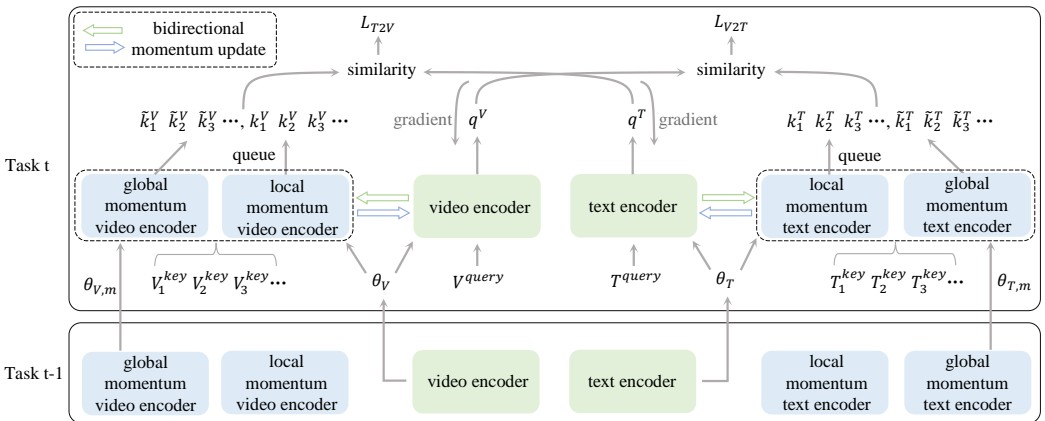

Figure 2: Schematic illustration of our BMU-MoCo. The momentum update strategy is applied to both encoders and momentum encoders (*i.e.*, bidirectional). To exploit earlier knowledge, we further maintain a pair of global momentum encoders with the same BMU strategy, whose parameters are inherited across tasks and only initialized at the very beginning.

where $m$ is the coefficient of momentum update. To form the contrastive learning loss of cross-modal MoCo, we need two consistent queues to preserve the negative video/text samples. In detail, the key embeddings $k_i^V$, $k_i^T$ of $V_i$, $T_i$ are firstly acquired by momentum video and text encoders:

$$k_i^V = f_{\theta_{V,m}}(V_i), \quad k_i^T = f_{\theta_{T,m}}(T_i). \tag{9}$$

We then respectively push $k_i^V$ and $k_i^T$ into the negative video queue $Q^V$ and the negative text queue $Q^T$ (after computing loss), where $Q^V = \{k_1^V, k_2^V, k_3^V, \cdots k_{N_Q}^V\}$ and $Q^T = \{k_1^T, k_2^T, k_3^T, \cdots, k_{N_Q}^T\}$ ($N_Q$ is the queue size). The contrastive losses of cross-modal MoCo (Base-MoCo) are:

$$\hat{L}_{V2T} = -\frac{1}{N_B} \sum_{i=1}^{N_B} \log \frac{\exp\left(\frac{q_i^V \cdot k_i^T}{\tau}\right)}{\exp\left(\frac{q_i^V \cdot k_i^T}{\tau}\right) + \sum_{j=1}^{N_Q} \exp\left(\frac{q_i^V \cdot k_j^T}{\tau}\right)}, \tag{10}$$

$$\hat{L}_{T2V} = -\frac{1}{N_B} \sum_{i=1}^{N_B} \log \frac{\exp\left(\frac{q_i^T \cdot k_i^V}{\tau}\right)}{\exp\left(\frac{q_i^T \cdot k_i^V}{\tau}\right) + \sum_{j=1}^{N_Q} \exp\left(\frac{q_i^T \cdot k_j^V}{\tau}\right)}, \tag{11}$$

where $\tau$ is the temperature. Note that the queue size $N_Q$ is decoupled from the batch size $N_B$. Therefore, it can take a large value for better representation of the data distribution.

**Bidirectional Momentum Update.** Although achieving great success with non-streaming data (*e.g.*, a single video-text dataset), the original cross-modal MoCo has difficulty in coping with the catastrophic forgetting problem under our CVLM setting. To overcome this difficulty, we propose a novel bidirectional momentum update (BMU) strategy for cross-modal MoCo to review the old knowledge retained in momentum encoders at each training step. Concretely, for video/text encoders $f_{\theta_V}, f_{\theta_T}$, in addition to the end-to-end update by back-propagation, we further update their parameters $\theta_V, \theta_T$ using the parameters $\theta_{V,m}, \theta_{T,m}$ of momentum encoders $f_{\theta_{V,m}}, f_{\theta_{T,m}}$ by momentum update:

$$\theta_V = \hat{m} \cdot \theta_V + (1 - \hat{m}) \cdot \theta_{V,m}, \quad \theta_T = \hat{m} \cdot \theta_T + (1 - \hat{m}) \cdot \theta_{T,m}, \tag{12}$$

where $\hat{m}$ is a momentum coefficient, and $\theta_{V,m}, \theta_{T,m}$ are simultaneously updated by Eq. (8). Together, Eq. (8) and Eq. (12) compose our BMU strategy. Note that the advantages of BMU lie in two aspects: (1) At the beginning of each new task $t$, $\theta_{V,m}$ and $\theta_{T,m}$ are respectively initialized by $\theta_V$ and $\theta_T$, which makes the knowledge of task $t-1$ be preserved. (2) During the training process, $\theta_{V,m}$ and $\theta_{T,m}$ are constantly and slowly updated by the momentum update strategy, which enables our model to review the old knowledge but without sacrificing the performance on new tasks.

**Global Momentum Encoders.** To further enhance our BMU-MoCo, we propose to maintain a pair of global momentum encoders which can preserve earlier knowledge. As shown in Figure 2, they are only initialized at the very beginning of the whole training process under our CVLM setting,

and their parameters are transmitted across tasks. Formally, let $f_{\tilde{\theta}_{V,m}}$ and $f_{\tilde{\theta}_{T,m}}$ denote the global momentum video and text encoders, respectively. Their parameters $\tilde{\theta}_{V,m}$ and $\tilde{\theta}_{T,m}$ are updated by the BMU strategy along with the parameters $\theta_V$ and $\theta_T$ of encoders:

$$\theta_V = \hat{m} \cdot \theta_V + (1 - \hat{m}) \cdot \tilde{\theta}_{V,m}, \quad \theta_T = \hat{m} \cdot \theta_T + (1 - \hat{m}) \cdot \tilde{\theta}_{T,m}, \tag{13}$$

$$\tilde{\theta}_{V,m} = m \cdot \tilde{\theta}_{V,m} + (1 - m) \cdot \theta_V, \quad \tilde{\theta}_{T,m} = m \cdot \tilde{\theta}_{T,m} + (1 - m) \cdot \theta_T. \tag{14}$$

Note that Eq. (12) and Eq. (13) are implemented subsequently. For each video-text input $\{V_i, T_i\}$, we obtain a new group of key embeddings $\tilde{k}_i^V$, $\tilde{k}_i^T$ with the global momentum encoders $f_{\tilde{\theta}_{V,m}}, f_{\tilde{\theta}_{T,m}}$:

$$\tilde{k}_i^V = f_{\tilde{\theta}_{V,m}}(V_i), \quad \tilde{k}_i^T = f_{\tilde{\theta}_{T,m}}(T_i). \tag{15}$$

We push $\tilde{k}_i^V$ and $\tilde{k}_i^T$ respectively into two negative queues $\tilde{Q}^V$ and $\tilde{Q}^T$, where $\tilde{Q}^V = \{\tilde{k}_1^V, \tilde{k}_2^V, \tilde{k}_3^V, \cdots, \tilde{k}_{N_Q}^V\}$, $\tilde{Q}^T = \{\tilde{k}_1^T, \tilde{k}_2^T, \tilde{k}_3^T, \cdots, \tilde{k}_{N_Q}^T\}$. Note that each query embedding (*e.g.*, $q_i^T$) has two corresponding positive embeddings ($k_i^V, \tilde{k}_i^V$) and two corresponding negative queues ($Q^V, \tilde{Q}^V$). The cross-modal contrastive losses are defined as:

$$L_{V2T} = -\frac{1}{N_B} \sum_{i=1}^{N_B} \log \frac{\exp\left(\frac{q_i^V \cdot k_i^T}{\tau}\right) + \exp\left(\frac{q_i^V \cdot \tilde{k}_i^T}{\tau}\right)}{\exp\left(\frac{q_i^V \cdot k_i^T}{\tau}\right) + \exp\left(\frac{q_i^V \cdot \tilde{k}_i^T}{\tau}\right) + \sum_{j=1}^{N_Q}\left[\exp\left(\frac{q_i^V \cdot k_j^T}{\tau}\right) + \exp\left(\frac{q_i^V \cdot \tilde{k}_j^T}{\tau}\right)\right]}, \tag{16}$$

$$L_{T2V} = -\frac{1}{N_B} \sum_{i=1}^{N_B} \log \frac{\exp\left(\frac{q_i^T \cdot k_i^V}{\tau}\right) + \exp\left(\frac{q_i^T \cdot \tilde{k}_i^V}{\tau}\right)}{\exp\left(\frac{q_i^T \cdot k_i^V}{\tau}\right) + \exp\left(\frac{q_i^T \cdot \tilde{k}_i^V}{\tau}\right) + \sum_{j=1}^{N_Q}\left[\exp\left(\frac{q_i^T \cdot k_j^V}{\tau}\right) + \exp\left(\frac{q_i^T \cdot \tilde{k}_j^V}{\tau}\right)\right]}, \tag{17}$$

where $\tau$ is the temperature. Now we have the final loss of BMU-MoCo for our CVLM setting:

$$L_{final} = L_{V2T} + L_{T2V}. \tag{18}$$

The full (pseudocode) algorithm of our BMU-MoCo is presented in the supplementary material.

## 4 Experiments

### 4.1 Experimental Setup

**Datasets.** Our CVLM setting is defined over a sequence of five video-text datasets: (1) VATEX [54] is a large-scale open-domain dataset, which has 25,991 videos with 250K text descriptions for training, 3,000 videos for validation and 6,000 videos for testing. (2) ActivityNet [25] is an action domain dataset, which consists of 20K YouTube videos with 100K text descriptions. We follow the standard setting in [4, 28] to use 10K videos for training and 4.9K for test (the val1 split), where all texts of each video are concatenated into one query paragraph. (3) MSR-VTT [55] contains 10K videos, with 20 text descriptions per video. We follow the 1k-A split in recent works [28, 4, 56, 33] with 9K training videos and 1K test videos. (4) DiDeMo [20] consists of 10K Flickr videos with 40K text annotations. Following [4, 28], we train and evaluate our model on paragraph-to-video retrieval (the same setting for ActivityNet). (5) MSVD [10] has 1,200 videos with 48K texts for training, 100 videos for validation and 670 ones for testing. Overall, there are around 50K videos with 500K text descriptions in all five datasets (*i.e.*, each dataset has 100K video-text pairs in average).

**Evaluation Metrics.** Similar to the standard video-language modeling setting, we evaluate the text-to-video retrieval performance of a model on Recall@1 (shortened as R@1). R@1 refers to the percentage of text queries that correctly retrieve the ground-truth candidate at top-1. For our CVLM setting, we further define two evaluation metrics: forgetting rate (FR) and harmonic mean (HM). Formally, let $\mathcal{M}_i$ denote the model after trained on task $i$ and $\mathcal{A}_t^i$ ($t \leq i$) denotes the R@1 result of $\mathcal{M}_i$ on task $t$. The overall R@1 ($\frac{1}{n}\sum_{t=1}^{n}\mathcal{A}_t^n$) is the average R@1 results of the final model $\mathcal{M}_n$ on all tasks. Based on these notations, we then define the FR and HM as follows:
(1) **Forgetting rate (FR)** of $\mathcal{M}_i$ on task $t$ ($t \leq i$) is the performance decrease between $\mathcal{M}_i$ and $\mathcal{M}_t$: FR $= \mathcal{A}_t^t - \mathcal{A}_t^i$, where a lower FR indicates the model forgets less knowledge. Note that there is no catastrophic forgetting (FR $= 0$) when $t = i$. The Overall FR is obtained by just summing the FR results of the final model $\mathcal{M}_n$ across all tasks: Overall FR $= \sum_{t=1}^{n}(\mathcal{A}_t^t - \mathcal{A}_t^n)$.

Table 1: Comparative results obtained by the final model $\mathcal{M}_n$ on five video-text datasets/tasks under our CVLM setting: VATEX [54] (*i.e.*, Task1), ActivityNet [25] (*i.e.*, Task2), MSR-VTT [55] (*i.e.*, Task3), DiDeMo [20] (*i.e.*, Task4), MSVD [10] (*i.e.*, Task5). For fair comparison, all baseline models are re-implemented based on the same cross-modal MoCo architecture for our CVLM setting. $^{\dagger}$ denotes applying extra encoders, including encoders from the last task (*e.g.*, LwF [32]) and global momentum encoders (*e.g.*, our BMU-MoCo). 'Mem.' denotes applying memory buffer during training. 'BMU-MoCo (local)' denotes BMU-MoCo without global momentum encoders.

| Method | Mem. | Task1 | | Task2 | | Task3 | | Task4 | | Task5 | Overall | | |
|---|---|---|---|---|---|---|---|---|---|---|---|---|---|
| | | R@1↑ | FR↓ | R@1↑ | FR↓ | R@1↑ | FR↓ | R@1↑ | FR↓ | R@1↑ | R@1↑ | FR↓ | HM↑ |
| Base-MoCo [35] | No | 38.99 | 15.30 | 18.61 | 15.23 | 28.00 | 5.60 | 28.22 | 7.27 | 40.28 | 30.82 | 43.40 | 34.67 |
| LwF$^{\dagger}$ [32] | No | 42.02 | 12.27 | 19.95 | 14.87 | 28.90 | 6.00 | 29.91 | 6.98 | 37.91 | 32.24 | 40.12 | 35.81 |
| ER-ring [9] | Yes | 41.99 | 12.30 | 22.09 | 11.79 | 29.80 | 5.40 | 30.31 | 5.08 | 38.53 | 32.54 | 34.57 | 35.67 |
| DER [5] | Yes | 40.15 | 14.14 | 21.35 | 12.65 | 28.80 | 5.00 | 30.71 | 4.09 | 39.96 | 32.19 | 35.88 | 35.39 |
| Co2L$^{\dagger}$ [6] | Yes | 41.23 | 13.06 | 21.74 | 13.06 | 27.50 | 5.30 | 30.41 | 5.38 | 39.29 | 31.58 | 34.48 | 35.06 |
| LUMP$^{\dagger}$ [37] | Yes | 40.16 | 14.13 | 21.78 | 12.37 | 30.50 | 3.00 | 29.91 | 4.99 | 39.39 | 32.45 | 34.49 | 35.56 |
| BMU-MoCo (local) | No | 46.82 | 7.47 | 23.27 | 10.84 | 30.00 | 3.40 | 31.21 | 4.08 | **41.94** | 34.65 | 25.79 | 37.05 |
| BMU-MoCo$^{\dagger}$ | No | **48.48** | **5.81** | **23.45** | 10.43 | **30.80** | **2.90** | **32.80** | **3.49** | 41.83 | **35.47** | **22.63** | **37.59** |

(2) **Harmonic mean (HM)** calculates the harmonic mean of the overall R@1 (current) and the overall R@1 (final), where the overall R@1 (current) denotes the average of the R@1 values obtained by each current model $\mathcal{M}_i$ ($i = 1, 2, \cdots, n$) on each current task $i$, and the overall R@1 (final) denotes the average R@1 for the final model $\mathcal{M}_n$ on all tasks. Formally, we have: HM $= \frac{2 \cdot \frac{1}{n} \sum_{i=1}^{n} \mathcal{A}_i^i \cdot \frac{1}{n} \sum_{i=1}^{n} \mathcal{A}_i^n}{\frac{1}{n} \sum_{i=1}^{n} \mathcal{A}_i^i + \frac{1}{n} \sum_{i=1}^{n} \mathcal{A}_i^n}$. Note that HM can alleviate the trade-off problem between overall R@1 (current) and overall R@1 (final), which is otherwise an inherent limitation of FR. Specifically, when a model has a lower overall R@1 (current) and a lower overall R@1 (final), it could also have a better/lower FR (which is unsatisfactory) but still lead to a worse/lower HM (see Figure 4(c)).

**Implementation Details.** Recent works on vision + language demonstrate that video-language models benefit from image-text pre-training [28, 4], which can accelerate the model convergence and is more suitable for the real-world application scenarios. We thus apply ViT-Base [13]/BERT-Base [12] as our image/text encoder, and follow the recent state-of-the-art MoCo-based model COTS [35] to pre-train our model with 5.3M image-text pairs. We then sequentially train all the models (BMU-MoCo and all competitors) on five video-text datasets/tasks. For each task, we train a model for 10 epochs and choose the best trained one w.r.t. the validation R@1 results. For those competitors using a memory buffer during training (*e.g.*, ER-ring [9]), we set the memory size to 10% of the average data size 100K (*i.e.*, 10K video-text pairs). Note that the percentage 10% is larger than the buffer size of most recent rehearsal-based continual learning methods [9, 5, 37]. More implementation details are given as follows: (1) In the training phase, all sampled frames of each video are resized to 384×384 and augmented by gray-scaling and color-jitter. (2) For the first epoch of each task under our CVLM setting, we set the learning rate to 5e-5 and decay it to 5e-6 afterwards. (3) We select the two momentum coefficients $m = 0.99$, $\hat{m} = 0.99$, and the temperature $\tau = 0.07$. We set the batch size $N_B$ to 48 and the queue size $N_Q$ to 1,440. (4) The total training time on five tasks is around 20 hours with 8 Tesla V100 GPUs for each model.

### 4.2 Main Results

Table 1 summarizes the comparative results in terms of text-to-video retrieval (R@1), forgetting rate and harmonic mean (HM) obtained by the final model $\mathcal{M}_n$ (per method) on five datasets. We re-implement five recent continual learning methods (fused with cross-modal MoCo) under our CVLM setting, including rehearsal-based methods (ER-ring [9], DER [5]), regularization-based methods (LwF [32]) and their combinations (Co2L [6], LUMP [37]). We can observe that: (1) Our BMU-MoCo outperforms recent methods by large margins without using any extra memory or dynamically extended networks. Concretely, our method achieves the best R@1 and FR results on all tasks, and outperforms the second best by $2.93\%$ for overall R@1, $11.85\%$ for overall FR and $1.78\%$ for overall HM. (2) Without applying global momentum encoders, our BMU-MoCo (local) also beats all competitors, directly showing the effectiveness of our BMU strategy. (3) The improvements over Base-MoCO obtained by utilizing knowledge distillation (*e.g.*, LwF [32]) or extra memory data (*e.g.*, Co2L [6]) are limited due to the lack of category information under our CVLM setting.

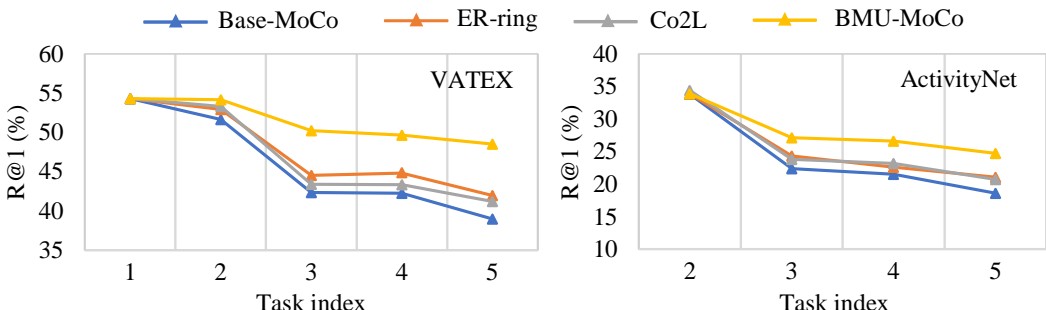

Figure 3: Detailed comparative results for text-to-video retrieval (R@1) obtained by each model $\mathcal{M}_i$ (per method) on the first two tasks: VATEX and ActivityNet. Note that the gap between the *beginning* and the *end* of each line denotes the forgetting rate.

Table 2: Ablation study results for our BMU-MoCo. 'Local' denotes applying the local momentum encoders, while 'Global' denotes applying the global momentum encoders. Results for text-to-video retrieval (R@1), forgetting rate (FR) and harmonic mean (HM) are reported.

| BMU | Local | Global | Task1 | | Task2 | | Task3 | | Task4 | | Task5 | Overall | | |
|---|---|---|---|---|---|---|---|---|---|---|---|---|---|---|
| | | | R@1↑ | FR↓ | R@1↑ | FR↓ | R@1↑ | FR↓ | R@1↑ | FR↓ | R@1↑ | R@1↑ | FR↓ | HM↑ |
| | | ✓ | 38.99 | 15.30 | 18.61 | 15.23 | 28.00 | 5.60 | 28.22 | 7.27 | 40.28 | 30.82 | 43.40 | 34.67 |
| | ✓ | | 38.95 | 15.97 | 18.51 | 15.45 | 30.10 | 4.60 | 28.91 | 6.78 | 39.89 | 31.47 | 42.75 | 35.16 |
| | ✓ | ✓ | 41.44 | 12.85 | 21.68 | 13.22 | 29.40 | 4.90 | 29.31 | 6.08 | 39.96 | 32.35 | 36.95 | 35.67 |
| ✓ | | ✓ | 46.82 | 7.47 | 23.27 | 10.84 | 30.00 | 3.40 | 31.21 | 4.08 | **41.94** | 34.65 | 25.79 | 37.05 |
| ✓ | ✓ | | 46.35 | 7.94 | 23.16 | 10.99 | 30.60 | 3.70 | 31.41 | 5.28 | 41.70 | 34.64 | 27.91 | 37.22 |
| ✓ | ✓ | ✓ | **48.48** | **5.81** | **23.45** | **10.43** | **30.80** | **2.90** | **32.80** | **3.49** | 41.83 | **35.47** | **22.63** | **37.59** |

Figure 3 shows more detailed comparative results for text-to-video retrieval (R@1) obtained by each model $\mathcal{M}_i$ (per method) on the first two datasets (VATEX [54] and ActivityNet [25]). We compare our BMU-MoCo with three representative competitors, including Base-MoCo [19], ER-ring [9], and Co2L [6]. Concretely, the left sub-figure presents the results of $\mathcal{M}_1 \sim \mathcal{M}_5$ (per method) on task 1 (VATEX), *i.e.*, $\mathcal{A}_1^i$ ($1 \leq i \leq 5$). The right sub-figure presents the results of $\mathcal{M}_2 \sim \mathcal{M}_5$ (per method) on task 2 (ActivityNet), *i.e.*, $\mathcal{A}_2^i$ ($2 \leq i \leq 5$). It can be observed that: (1) For task 1 (VATEX), the performance of our BMU-MoCo drops the most slowly after it is trained on the following tasks (task 2 to task 5). (2) For task 2 (ActivityNet), our BMU-MoCo also leads to the slowest performance drop after trained on the following tasks (task 3 to task 5). Overall, our BMU-MoCo indeed significantly alleviates the performance decrease problem during the whole training process.

### 4.3 Ablation Study

We first conduct ablation study to demonstrate the contributions of the BMU strategy, the local momentum encoders and the global momentum encoders applied in our BMU-MoCo. The ablative results are shown in Table 2. It can be clearly seen that: (1) With our BMU strategy, our model achieves remarkable improvements (4th row vs. 1st row). (2) Simultaneously applying local and global encoders is better than using only one of them (3rd row vs. 1st/2nd row), which indicates that the knowledge preserved in local and global momentum encoders are quite different and thus complementary to each other. (3) Our BMU strategy helps our model to excavate old knowledge preserved in different momentum encoders (6th row vs. 3rd row) and achieve the best performance (6th row vs. 4th/5th row), which further validates the effectiveness of our BMU.

Considering the core role of BMU, we thus analyze the impact of the momentum coefficient $\hat{m}$ utilized in our BMU-MoCo. According to COTS [35], the other momentum coefficient $m$ of our model is fixed at 0.99 (only the value of $\hat{m}$ is changed). Figure 4 shows the results for overall R@1 (current), overall R@1 (final), and overall HM, respectively. We find that the value of $\hat{m}$ cannot be too big or too small. Concretely, when $\hat{m}$ is too big (*e.g.*, 0.999 and 1), the knowledge preserved in momentum encoders cannot be well-reviewed by our model. When $\hat{m}$ is too small (*e.g.*, 0.9), the end-to-end update by back-propagation is influenced too much, which leads to bad results for

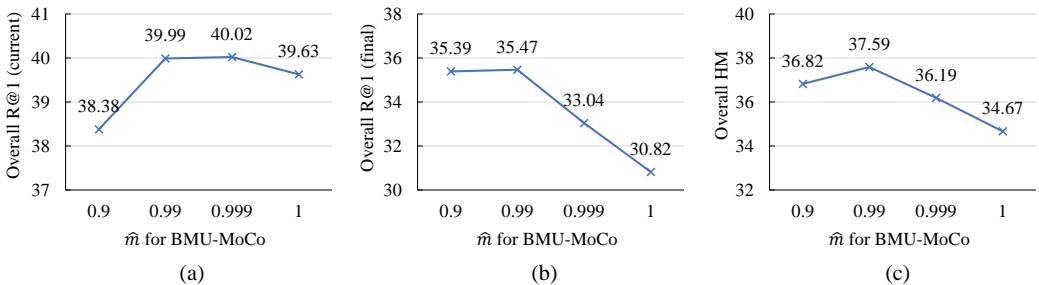

(a)             (b)             (c)

Figure 4: Comparative results of our BMU-MoCo with different momentum coefficient $\hat{m}$. The results for overall R@1 (current), overall R@1 (final) and overall HM are reported in (a–c), respectively.

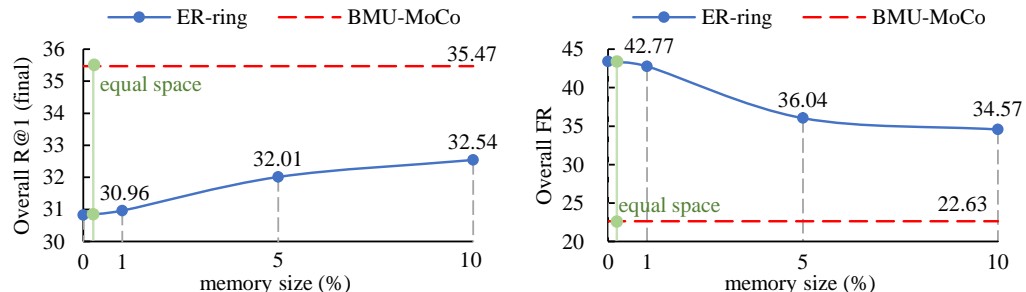

Figure 5: Comparative results obtained by different memory size. The red dotted line denotes our BMU-MoCo (with no memory) while the blue line denotes the representative rehearsal-based method ER-ring. The green line suggests that the storage space consumption of our BMU-MoCo (with global momentum encoders) is equal to ER-ring (with 0.05% memory to store videos).

overall R@1 (current) and overall HM. It is worth mentioning that the model with smaller $\hat{m}$ (0.9) has lower/better FR (14.95) since it sacrifices the model performance on overall R@1 (current). This phenomenon demonstrates the necessity of utilizing the overall HM to measure the overall (trade-off) model performance. Therefore, we set the momentum coefficient $\hat{m}$ to 0.99 in all our experiments, which helps our model to review old knowledge while learning well on new tasks.

## 4.4 Further Evaluation

To demonstrate both the efficiency and effectiveness of our BMU-MoCo under our CVLM setting, we compare our model with a representative rehearsal-based method ER-ring [9] by different memory sizes in Figure 5. Note that our BMU-MoCo has two global momentum encoders that need 0.5GB more storage space than the original cross-modal MoCo (used by all competitors including ER-ring). As shown in Figure 5, when the memory size of ER-ring becomes 0.05%, it equals to the size of extra storage space used by our BMU-MoCo (but our model performs significantly better). In real-world application scenarios, the memory size of rehearsal-based methods like ER-ring enlarges rapidly as the number of tasks grows, while the fixed extra space size (0.5GB) of our BMU-MoCo is negligible. More importantly, our BMU-MoCo (with a fixed 0.5GB sapce size) even outperforms ER-ring using 10% memory (about 200GB under our CVLM setting) by large margins on both overall R@1 (final) and overall FR. This directly indicates the efficiency and effectiveness of our BMU-MoCo.

Note that the key factor of our BMU-MoCo is to learn holistic knowledge from momentum encoders by momentum update. This momentum update strategy has two characteristics: (1) encoders are updated on each iteration by momentum update and (2) momentum encoders are simultaneously updated. We thus consider two alternative strategies to further validate the effectiveness of our BMU-MoCo in Table 3: (1) BMU-MoCo-ensemble updates the encoders by adding the parameters of the momentum encoders at the end of each task (*i.e.*, ensemble), which is different from the *iteration-by-iteration* strategy of our BMU-MoCo. It can be observed that BMU-MoCo-ensemble learns better than the baseline method (Base-MoCo) but still has a large performance gap compared with our BMU-MoCo. (2) BMU-MoCo-fixed fixes the momentum encoders during each task (only re-initialized by encoders at the beginning of each task) and utilizes it to update the encoders in

Table 3: Different momentum update strategies for our BMU-MoCo. Results for text-to-video retrieval (R@1), forgetting rate (FR) and harmonic mean (HM) are reported.

| Method | Task1 | | Task2 | | Task3 | | Task4 | | Task5 | Overall | | |
|---|---|---|---|---|---|---|---|---|---|---|---|---|
| | R@1↑ | FR↓ | R@1↑ | FR↓ | R@1↑ | FR↓ | R@1↑ | FR↓ | R@1↑ | R@1↑ | FR↓ | HM↑ |
| Base-MoCo | 38.99 | 15.30 | 18.61 | 15.23 | 28.00 | 5.60 | 28.22 | 7.27 | 40.28 | 30.82 | 43.40 | 34.67 |
| BMU-MoCo-ensemble | 41.44 | 12.85 | 21.68 | 13.22 | 29.40 | 4.90 | 29.31 | 6.08 | 39.96 | 32.35 | 36.95 | 35.67 |
| BMU-MoCo-fixed | **51.36** | **2.93** | 23.12 | **7.38** | 25.10 | **1.50** | 30.11 | **2.00** | 38.19 | 34.07 | **13.81** | 35.12 |
| BMU-MoCo | 48.48 | 5.81 | **23.45** | 10.43 | **30.80** | 2.90 | **32.80** | 3.49 | **41.83** | **35.47** | 22.63 | **37.59** |

Table 4: Comparative results of unsupervised continual learning (UCL). "Multi-Task" represents the upper-bound method which is based on multi-task learning (over all tasks). "Finetune" is the lower-bound method that directly fine-tunes the model across all tasks without other strategies.

| Method | Split CIFAR-10 | | Split CIFAR-100 | | Split Tiny-ImageNet | |
|---|---|---|---|---|---|---|
| | Accuracy↑ | Forgetting↓ | Accuracy↑ | Forgetting↓ | Accuracy↑ | Forgetting↓ |
| Multi-Task | 95.76 | – | 86.31 | – | 82.89 | – |
| Finetune | 90.11 | 5.42 | 75.42 | 10.19 | 71.07 | 9.48 |
| PNN [48] | 90.93 | – | 66.58 | – | 62.15 | – |
| SI [60] | 92.75 | 1.81 | 80.08 | 5.54 | 72.34 | 8.26 |
| DER [5] | 91.22 | 4.63 | 77.27 | 9.31 | 71.90 | 8.36 |
| LUMP [37] | 91.00 | 2.92 | 82.30 | 4.17 | 76.66 | 3.54 |
| BMU-MoCo | **92.80** | **1.98** | **83.81** | **3.69** | **77.45** | **3.39** |

the same iteration-by-iteration way of our BMU-MoCo. It can be observed that BMU-MoCo-fixed focuses on reviewing the old knowledge but sacrifices the model performance on current tasks, which leads to worse overall R@1 (final) and HM. Therefore, considering all evaluation metrics (R@1, FR, and HM), the update strategy (for encoders) of our BMU-MoCo is better than its alternatives.

Finally, to show the transferability of our BMU-MoCo under other continual learning settings, we apply it to the unsupervised continual learning (UCL) setting which is originally proposed in LUMP [37]. This UCL setting includes three benchmark datasets for training and evaluation: (1) Split CIFAR-10 [26] consists of five tasks with 2 random classes per task; (2) Split CIFAR-100 [26] consists of 20 tasks with 5 random classes per task; (3) Split Tiny-ImageNet consists of 20 tasks with 5 random classes (out of 100 classes from ImageNet [11]) per task. The comparative results of UCL are shown in Table 4. We can observe that our BMU-MoCo outperforms all the competitors including PNN [48], SI [60], DER [5], and LUMP [37], showing the effectiveness of our BMU-MoCo under the UCL setting. This also demonstrates the general applicability of our BMU strategy.

## 5 Conclusion

In this paper, we propose a new continual video-language modeling (CVLM) setting, where models are supposed to be sequentially trained on five widely-used video-text datasets. To overcome the catastrophic forgetting and heavy resource consumption challenges, we propose a novel framework BMU-MoCo, which is a cross-modal MoCo-based model with bidirectional momentum update (BMU). We maintain both local and global momentum encoders with our BMU strategy to review broader old knowledge while learning on new tasks. Extensive experimental results show that our BMU-MoCo outperforms recent competitors by large margins, even without using extra memory data or dynamically extended networks. The limitation of our work lies in that we have only evaluated BMU-MoCo under the CVLM setting, and thus we need to transfer it to other continual learning settings (*e.g.*, continual image-text pre-training) for comprehensive study.

## Acknowledgments and Disclosure of Funding

This work was supported in part by National Natural Science Foundation of China (61976220 and 61832017), Beijing Outstanding Young Scientist Program (BJJWZYJH012019100020098), and the Research Seed Funds of School of Interdisciplinary Studies, Renmin University of China.

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
