# BMU-MoCo: Bidirectional Momentum Update for Continual Video-Language Modeling – Supplementary Material –

**Yizhao Gao**[1,2] **Nanyi Fei**[1,2] **Haoyu Lu**[1,2] **Zhiwu Lu**[1,2,*] **Hao Jiang**[3] **Yijie Li**[3] **Zhao Cao**[3]

[1]Gaoling School of Artificial Intelligence, Renmin University of China, Beijing, China
[2]Beijing Key Laboratory of Big Data Management and Analysis Methods
[3]Huawei Poisson Lab, Hangzhou, Zhejiang, China
{gaoyizhao, luzhiwu}@ruc.edu.cn

## 1 Full Algorithm of BMU-MoCo

We provide the pseudocode of our BMU-MoCo in Algorithm 1.

---

**Algorithm 1** Pseudocode of BMU-MoCo.

---

**Input:** Video encoder $f_{\theta_V}$ (with parameters $\theta_V$);
       Text encoder $f_{\theta_T}$ (with parameters $\theta_T$);
       A dataset $\mathcal{D} = [\mathcal{D}_1, \mathcal{D}_2, \cdots, \mathcal{D}_n]$ of $n$ video-text tasks;
       The hyper-parameters $m, \hat{m}, \tau$.
**Output:** The learned $f_{\theta_V}^*$ and $f_{\theta_T}^*$
1: Initialize the local momentum encoders $f_{\theta_{V,m}} = f_{\theta_V}, f_{\theta_{T,m}} = f_{\theta_T}$;
2: Initialize the global momentum encoders $f_{\tilde{\theta}_{V,m}} = f_{\theta_V}, f_{\tilde{\theta}_{T,m}} = f_{\theta_T}$;
3: Randomly initialize queues $Q^V, Q^T, \tilde{Q}^V, \tilde{Q}^T$;
4: **for all** task = 1, 2, $\cdots$, n **do**
5:    **for all** iteration = 1, 2, $\cdots$, MaxIteration **do**
6:       Sample a mini-batch with $N_B$ video-text pairs $\{V_i, T_i\}_{i=1}^{N_B}$ from $\mathcal{D}_t$;
7:       Obtain the query embeddings $q_i^V, q_i^T$ with Eq. (7);
8:       Obtain the local key embeddings $k_i^V, k_i^T$ with Eq. (9);
9:       Obtain the global key embeddings $\tilde{k}_i^V, \tilde{k}_i^T$ with Eq. (15);
10:      Compute the final loss $L_{final}$ with Eqs. (16)-(18);
11:      Compute the gradients $\nabla_{f_{\theta_V}} L_{final}$ and $\nabla_{f_{\theta_T}} L_{final}$ with Eq. (18);
12:      Update $f_{\theta_V}$ and $f_{\theta_T}$ using Adam;
13:      Update $f_{\theta_V}$ and $f_{\theta_T}$ with Eq. (12) and Eq. (13);
14:      Update $f_{\theta_{V,m}}$ and $f_{\theta_{T,m}}$ with Eq. (8);
15:      Update $f_{\tilde{\theta}_{V,m}}$ and $f_{\tilde{\theta}_{T,m}}$ with Eq. (14);
16:      Enqueue $k_i^V, k_i^T$ to $Q^V, Q^T$,;
17:      Enqueue $\tilde{k}_i^V, \tilde{k}_i^T$ to $\tilde{Q}^V, \tilde{Q}^T$;
18:      Dequeue the earliest embeddings from $Q^V, Q^T, \tilde{Q}^V, \tilde{Q}^T$;
19:    **end for**
20:    Re-initialize the local momentum encoders $f_{\theta_{V,m}} = f_{\theta_V}, f_{\theta_{T,m}} = f_{\theta_T}$;
21: **end for**
22: **return** the found best $f_{\theta_V}^*$ and $f_{\theta_T}^*$.

---

---
*The corresponding author.

36th Conference on Neural Information Processing Systems (NeurIPS 2022).

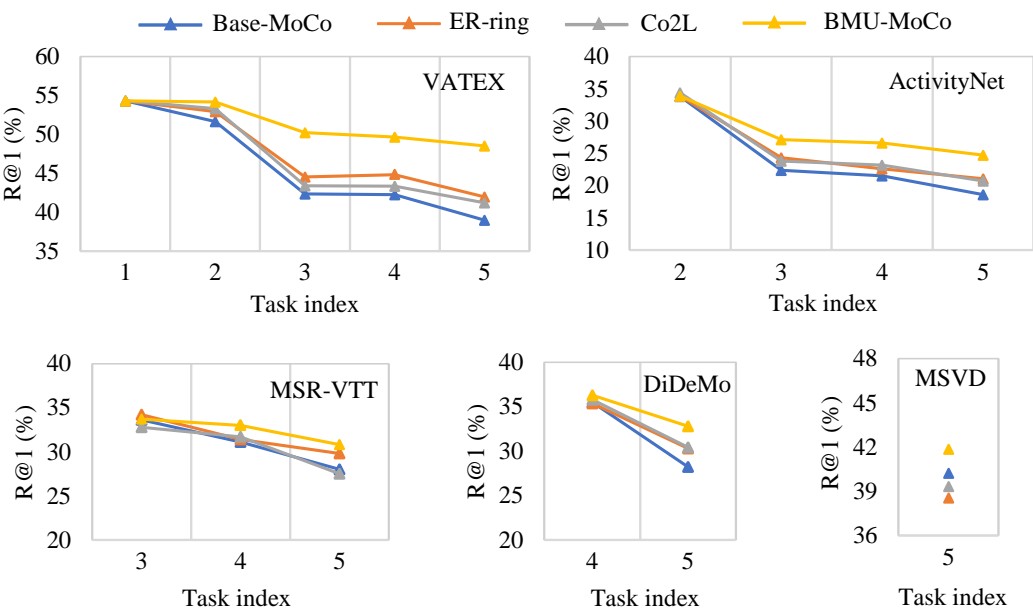

Figure 1: Detailed comparative results for text-to-video retrieval (R@1) obtained by each model $\mathcal{M}_i$ (per method) on all five tasks.

Table 1: Comparative results obtained by the final model $\mathcal{M}_n$ on five video-text datasets/tasks under our CVLM setting. $^\dagger$ denotes applying extra encoders. 'Mem.' denotes applying memory buffer during training. 'BMU-MoCo (local)' denotes BMU-MoCo without global momentum encoders. The R@5 results and its corresponding FR/HM are reported.

| Method | Mem. | Task1 R@5↑ | Task1 FR↓ | Task2 R@5↑ | Task2 FR↓ | Task3 R@5↑ | Task3 FR↓ | Task4 R@5↑ | Task4 FR↓ | Task5 R@5↑ | Overall R@5↑ | Overall FR↓ | Overall HM↑ |
|---|---|---|---|---|---|---|---|---|---|---|---|---|---|
| Base-MoCo [9] | No | 70.93 | 14.72 | 44.62 | 21.42 | 56.50 | 5.70 | 55.73 | 9.57 | 70.36 | 59.63 | 51.41 | 64.36 |
| LwF$^\dagger$ [8] | No | 74.23 | 11.42 | 46.98 | 18.65 | 55.10 | 4.90 | 56.83 | 8.17 | 71.01 | 60.83 | 43.14 | 64.86 |
| ER-ring [3] | Yes | 75.13 | 10.52 | 48.77 | 16.37 | 56.00 | 5.50 | 59.72 | 4.79 | 68.79 | 61.68 | 37.18 | 65.19 |
| DER [1] | Yes | 72.01 | 13.64 | 47.53 | 17.59 | **57.10** | 4.20 | 59.52 | 3.99 | 70.12 | 61.26 | 39.42 | 64.96 |
| Co2L$^\dagger$ [2] | Yes | 73.33 | 12.32 | 47.90 | 17.87 | 55.90 | 3.60 | 58.13 | 6.58 | 69.72 | 60.99 | 40.47 | 64.77 |
| LUMP$^\dagger$ [10] | Yes | 73.39 | 12.26 | 46.96 | 18.28 | 56.70 | 5.30 | 60.92 | **3.59** | 69.96 | 61.59 | 39.43 | 65.29 |
| BMU-MoCo (local) | No | 79.83 | 5.82 | **52.31** | 13.36 | 56.70 | 2.60 | 59.92 | 4.99 | **72.49** | 64.25 | 26.77 | 66.82 |
| BMU-MoCo$^\dagger$ | No | **81.39** | **4.26** | 52.21 | **12.54** | 56.70 | **2.20** | **61.02** | 4.18 | 72.17 | **64.69** | **23.18** | **66.93** |

## 2 More Comparative Results

**Detailed R@1 Results.** In our main paper, we present the detailed results of the first two tasks (VATEX [11] and ActivityNet [7]) in Figure 3 for simplicity. Here, we further provide the detailed comparative results of other tasks (MSR-VTT [12], DiDeMo [6], and MSVD [4]) in Figure 1. Concretely, for each sub-figure (with task index $t$), we present the results of $\mathcal{M}_t \sim \mathcal{M}_5$ on task $t$, i.e., $\mathcal{A}_t^i$ ($t \leq i \leq 5$). We observe that our BMU-MoCo outperforms its competitors on all tasks and leads to the slowest performance degradation. Moreover, our BMU-MoCo simultaneously ensures that the model achieves competitive performance on current tasks (the beginning of each line).

**R@5 and R@10 Results.** For comprehensive study, we evaluate our BMU-MoCo and its competitors with the widely-used metrics Recall@5 (R@5) and Recall@10 (R@10) in Table 1 and Table 2, respectively. Results for the forgetting rate (FR) and harmonic mean (HM) are also re-calculated based on R@5 or R@10. We can see that our BMU-MoCo outperforms all competitors by a large margin, especially on the overall metrics (R@5/R@10, FR, and HM).

**Visualization of Training Loss.** To show that our re-implemented infoNCE loss (i.e., Eq. (16) of our main paper) is stable, we respectively draw two training loss plots on ActivityNet and MSR-VTT

Table 2: Comparative results obtained by the final model $\mathcal{M}_n$ on five video-text datasets/tasks under our CVLM setting. $\dagger$ denotes applying extra encoders. 'Mem.' denotes applying memory buffer during training. 'BMU-MoCo (local)' denotes BMU-MoCo without global momentum encoders. The R@10 results and its corresponding FR/HM are reported.

| Method | Mem. | Task1 R@10↑ | FR↓ | Task2 R@10↑ | FR↓ | Task3 R@10↑ | FR↓ | Task4 R@10↑ | FR↓ | Task5 R@10↑ | Overall R@10↑ | FR↓ | HM↑ |
|---|---|---|---|---|---|---|---|---|---|---|---|---|---|
| Base-MoCo [9] | No | 81.33 | 10.92 | 59.14 | 20.34 | 65.70 | 6.90 | 66.30 | 8.18 | 80.14 | 70.52 | 46.34 | 74.87 |
| LwF$^\dagger$ [8] | No | 84.10 | 8.15 | 61.38 | 17.48 | 65.40 | 5.80 | 65.80 | 9.27 | 80.20 | 71.38 | 40.65 | 75.32 |
| ER-ring [3] | Yes | 84.43 | 8.09 | 63.43 | 15.03 | 66.30 | 4.20 | 70.59 | 3.59 | 79.11 | 72.77 | 30.91 | 75.73 |
| DER [1] | Yes | 81.76 | 10.49 | 61.62 | 16.80 | 67.80 | 3.20 | 70.79 | 3.79 | 79.62 | 72.32 | 34.28 | 75.59 |
| Co2L$^\dagger$ [2] | Yes | 82.97 | 9.28 | 61.48 | 17.53 | 66.40 | 3.60 | 68.59 | 6.98 | 79.82 | 71.85 | 37.39 | 75.40 |
| LUMP$^\dagger$ [10] | Yes | 82.99 | 9.26 | 60.89 | 17.80 | 67.20 | 2.30 | 70.49 | 4.78 | 79.23 | 72.16 | 34.14 | 75.42 |
| BMU-MoCo (local) | No | 88.24 | 4.01 | **66.79** | 11.94 | **69.00** | 1.20 | 69.79 | 5.09 | 81.99 | 75.16 | 22.24 | 77.32 |
| BMU-MoCo$^\dagger$ | No | **89.36** | **2.89** | **66.79** | **11.16** | 67.50 | 1.80 | **71.88** | **3.19** | **82.09** | **75.52** | **19.04** | **77.37** |

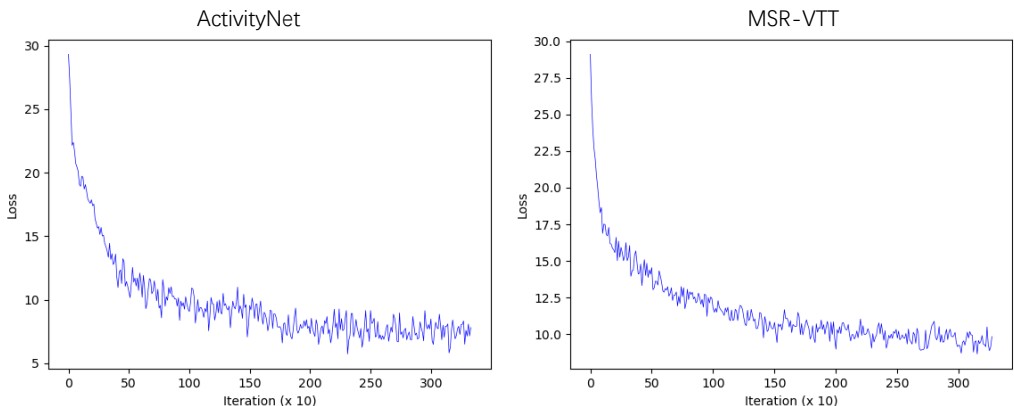

Figure 2: The change of training loss on ActivityNet and MSR-VTT.

in Figure 2. We can see that the losses tend to converge after 3,000 training iterations. Therefore, our re-implemented infoNCE loss is indeed effective and stable.

## 3 Implementation Details of Baseline Methods

In Table 1 of our main paper, we compare our BMU-MoCo with many recent continual learning methods under our proposed CVLM setting. Since all the competitors are originally proposed for other continual learning settings (e.g., unsupervised continual learning for image classification), we thus re-implement them to adapt to our proposed CVLM setting. Below we present the implementation details for each baseline method:

(1) LwF [8] is a regularization-based method which aligns the representations of old and current models as new data arrives. Under the CVLM setting, based on Base-MoCo, we additionally maintain a pair of video and text encoders whose parameters are copied from previous encoders. For each iteration, we align both video and text representations of old and current models.

(2) ER-ring [3] is a rehearsal-based method which has a ring-buffer to save the memory data. Following its continual learning strategy, we maintain a ring-buffer (with 10K capacity in total) to save the video-text pairs. The memory data are simply used as training samples in the training process. The model architecture is exactly the same as Base-MoCo.

(3) DER [1] is a rehearsal- and regularization-based method which aligns the logits of old and current models on the memory data. Note that the ring-buffer maintained for DER actually saves the retrieval logits obtained from the training process, instead of the original video-text pairs.

(4) Co2L [2] is also a rehearsal- and regularization-based method which saves the memory data and aligns the logits of old and current models on the new/old data. Different from DER, the logits

used in Co2L are obtained by different augmentations. Therefore, for each mini-batch, we take two augmentations to construct the alignment loss defined in Co2L.

(5) LUMP [10] is a rehearsal-based method which mixes up the memory data and the current data during training. Note that LUMP is originally based on SimSiam [5]. Under the CVLM setting, we transfer LUMP to cross-modal MoCo by taking the mixed data as positive samples while pushing them into negative queues after loss calculation.