# OpenReview forum: "BMU-MoCo: Bidirectional Momentum Update for Continual Video-Language Modeling"
_NeurIPS.cc/2022/Conference — NeurIPS 2022 Accept_

### Official Review · Reviewer_oj9S · 2022-07-09

**Rating:** 5
**Confidence:** 4
**Soundness:** 2 fair
**Presentation:** 1 poor
**Contribution:** 2 fair

**Summary:**

The authors introduce a new setting of the Video-Language Modeling task in a continual learning scenario. They propose an evaluation schema based on text2video (cross-modal) retrieval with 5 different tasks which correspond to separate popular video-language modeling datasets. Moreover, the authors propose their own method, namely BMU-MoCo, based on MoCo self-supervised representation learning approach, and compare it against other continual learning methods. The proposed method gives the best results given the evaluation metrics that measure the performance degradation between consecutive tasks, particularly ‘Forgetting rate’ and ‘Harmonic Mean’.

**Questions:**

In Figure 1a) what do colors represent? What are the current models and the final model?

Could you explain in detail what happens when the queue size Nq is larger than Nb?

What does FR for Task 1 indicate? Is there a Task 0 then? The comparison is not clear.

Is 0.5 additional GB of memory for BMU-MoCo local only or both?

How are frames sampled and fed to ViT and how is the averaging over the whole video being performed?


**Limitations:**

The authors very briefly mention the limitations of their work, with no discussion of potentially negative impact. In my opinion, the limitations of this work are two-fold. First, as the authors mention, they only tackle the CVML task, however, to fully address this task, the results of state-of-the-art approaches on particular datasets should also be included, showing that they indeed struggle with catastrophic forgetting. Otherwise, it would be beneficial to address other cross-modal tasks.

**Strengths And Weaknesses:**

Strengths:
1.	The Paper is well-structured, and the contribution is clearly stated and supported in the experimental section.
2.	The authors propose a new approach to evaluate the effectiveness of VLM models in a continual learning setup with new metrics to evaluate the task.
3.	The proposed method gives the best results among other continual learning-based approaches.

Weaknesses:
1.	The method section is a bit messy and becomes hard to follow.
2.	Notations in 3. are definitely over complex, and could be simplified, e.g. by dropping ‘i’ in Eq. 1-6.
3.	The method is not entirely free from additional resources contrary to what the authors claim, since it requires an additional model (momentum encoder- local + global) to be trained and stored, as well as the queue for the training.
4.	Some typos and minor editing issues.
5.	The conclusions on the m hyperparameter are unclear.
6.	Both figures 1 are non-informative and confuse the reader.
7.	Lack of clear motivation why CVML among other cross-modal tasks and evaluation only on one of them seems to be quite limited.

---

> ### Author Response · Authors · 2022-08-01
> **Response to Reviewer oj9S - Part I**
>
> Thank you for the constructive comments and suggestions.
>
> **Q1. Weakness 1, 2, 4: The method section is a bit messy and becomes hard to follow. Notations in Sec. 3 are definitely over complex, and could be simplified. Some typos and minor editing issues.** \
> **A1.** Thanks. We have carefully polished our paper.
>
> **Q2. Weakness 3: The method is not entirely free from additional resources contrary to what the authors claim, since it requires an additional model (momentum encoder- local + global) to be trained and stored, as well as the queue for the training.** \
> **A2.** There seem to exist some misunderstandings. Note that the traditional MoCo [a] and its cross-modal versions (COTS [32] and HiT [30]) all utilize momentum encoders and queues to construct the contrastive learning objectives. In fact, it has been clearly claimed in MoCo that using momentum encoders and queues can greatly reduce the computational cost during training, since it can adopt a small batch size while still maintaining a large queue of negative samples (which is essential in contrastive learning). In this paper, our BMU-MoCo and all the competitors are based on the same basic Base-MoCo with momentum encoders and queues. Under such fair setting, we evaluate our BMU-MoCo by comparing it to all the competitors. Specifically, we have proposed two BMU-MoCo models, one only utilizes local momentum encoders and the other utilizes local+global momentum encoders: (1) For the former BMU-MoCo (local), it has already outperformed all the competitors with exactly the same architecture of Base-MoCo (i.e., without using any extra memory and dynamic networks). (2) For the latter BMU-MoCo (local+global), although it maintains more momentum encoders and queues than BMU-MoCo (local), the additional cost is limited (0.5 GB in total) and fixed (as the task number grows) while achieving better performance. In conclusion, our BMU-MoCo (local) beats all the competitors under a fair setting and our BMU-MoCo (local+global) further brings performance boost with limited extra cost.
>
> **Q3. Weakness 5: The conclusions on the m hyperparameter are unclear.** \
> **A3.** Thanks for pointing this out. Since the hyper-parameter m is for the basic MoCo architecture, we directly set m to 0.99 for all methods (the same m is used in COTS [32]), which has been stated in Lines 269--270 of our main paper. In addition, the effect of the hyper-parameter $\hat{m}$ has been clearly analyzed in Lines 268--279 of our main paper.
>
> **Q4. Weakness 6 and Question 1: Both figures 1 are non-informative and confuse the reader. In Figure 1a) what do colors represent? What are the current models and the final model?** \
> **A4.** Sorry for the confusion. Note that we have explained the concept of current models and final model in Lines 36--38 of our main paper. To be more specific, the CVLM setting has a sequence of 5 tasks and the models are supposed to be sequentially trained on all these tasks. Therefore, the result of the current model on Task i is obtain by evaluating the model on Task i right after trained on Task i (before it is trained on Task i+1); the result of the final model on Task i is obtain by evaluating the model on Task i after trained on all 5 tasks. Particularly, the results of the current and final models on Task 1 in Figure 1(a) show that the performance of Base-MoCo (on Task 1) drops significantly after trained on all 5 tasks.
>
> **Q5. Weakness 7: Lack of clear motivation why CVLM among other cross-modal tasks and evaluation only on one of them seems to be quite limited.** \
> **A5.** It is worth noting that our proposed BMU-MoCo is indeed generalizable to other cross-modal tasks (e.g., image-text pre-training setting), since the two-stream architecture is generic for these tasks. However, considering the resource consumption and the paper conciseness, we choose to study one task in this work, i.e., video language pre-training with streaming data, which has important practical significance but has not been discussed in earlier works. To that end, we propose a challenging/novel Continual Video Language Modeling (CVLM) setting and conduct extensive experiments on five video-text datasets to verify the effectiveness of our proposed BMU-MoCo. More importantly, our BMU-MoCo could be easily transferred to other continual learning settings, such as unsupervised continual learning (see our response to Q1 of Reviewer 6RoA) and continual reinforcement learning (see our response to Q1 of Reviewer ZHhB).
>
> [a] He, K., Fan, H., Wu, Y., Xie, S., and Girshick, R. Momentum contrast for unsupervised visual representation learning. CVPR 2020.

---

> > ### Comment · Reviewer_oj9S · 2022-08-08
> > **Thank you.**
> >
> > Thank you for providing all the due responses to my questions and concerns. Given that you addressed most of my remarks, I revised my score.

---

> > > ### Author Response · Authors · 2022-08-08
> > > **Further response**
> > >
> > > Thank you very much. We will carefully improve the writing in the final version.

---

> ### Author Response · Authors · 2022-08-01
> **Response to Reviewer oj9S - Part II**
>
> **Q6. Question 2: Could you explain in detail what happens when the queue size Nq is larger than Nb?** \
> **A6.** The queues used in our BMU-MoCo are the same as those in MoCo [a] (and cross-modal MoCo). Typically, the queue size is set to be much larger than the batch size to save a large quantity of negative samples. Concretely, after trained on each mini-batch with the batch size $N_b$, the extracted features are pushed into the queues (while the earliest batches are popped out) and the features stored in the queues are used as negative samples for contrastive learning. Please see MoCo [a] for more details.
>
> **Q7. Question 3: What does FR for Task 1 indicate? Is there a Task 0 then? The comparison is not clear.** \
> **A7.** We have defined the Forgetting Rate (FR) in Lines 209--212 of our main paper. Note that the results in Table 1 are obtained by the final model $M_5$. Therefore, according to our definition, the FR for Task 1 is the performance degradation on Task 1 when the model is trained after all 5 tasks (i.e., $A_1^1 - A_1^5$).
>
> **Q8. Question 4: Is 0.5 additional GB of memory for BMU-MoCo local only or both?** \
> **A8.** Sorry for the confusion. 0.5 GB is only for our full BMU-MoCo (local+global), which represents the additional memory for saving global momentum encoders. It becomes 0 GB for our BMU-MoCo (local), since all methods are implemented based on the same architecture (Base-MoCo).
>
> **Q9. Question 5: How are frames sampled and fed to ViT and how is the averaging over the whole video being performed?** \
> **A9.** Frames are randomly and uniformly sampled (8 frames per video), which is widely-used in recent video-language modeling works (e.g., ClipBERT [b] and Frozen [c]). After extracted all frame features, we simply average them to obtain the whole video features (see Section 3.2).
>
> **Q10. Limitations: In my opinion, the limitations of this work are two-fold. First, as the authors mention, they only tackle the CVML task, however, to fully address this task, the results of state-of-the-art approaches on particular datasets should also be included, showing that they indeed struggle with catastrophic forgetting. Otherwise, it would be beneficial to address other cross-modal tasks.** \
> **A10.** In this work, we choose to study the CVLM setting based on cross-modal MoCo, and the results in Figure 1 show that the catastrophic forgetting problem indeed exists. Since the state-of-the-art approaches to VLM including COTS [32] and HiT [30] have similar cross-modal MoCo architectures, they would also suffer from catastrophic forgetting. Therefore, our study on the CVLM setting is vital for video-language modeling with streaming data. Additionally, our proposed BMU-MoCo is generalizable and can be transferred to other cross-modal tasks or other continual learning settings (see our response to Q5).
>
> We wish that our response has addressed your concerns, and turns your assessment to the positive side. If you have any questions, please feel free to let us know during the rebuttal window. We appreciate your suggestions and comments! Thank you!
>
> [a] He, K., Fan, H., Wu, Y., Xie, S., and Girshick, R. Momentum contrast for unsupervised visual representation learning. CVPR 2020.\
> [b] Lei, J., Li, L., Zhou, L., Gan, Z., Berg, T. L., Bansal, M., and Liu, J., Less is more: ClipBERT for video-and-language learning via sparse sampling, CVPR 2021.\
> [c] Bain, M., Nagrani, A., Varol, G., and Zisserman, A., Frozen in time: A joint video and image encoder for end-to-end retrieval, ICCV 2021.

---

> ### Author Response · Authors · 2022-08-07
> **Looking forward to your rebuttal discussion**
>
> Dear Reviewer oj9S,
>
> Thanks again for your insightful suggestions and comments. As the deadline for discussion is approaching, we are happy to provide any additional clarifications that you may need.
>
> In our previous response, we have carefully studied your comments and provided new experiments/additional explanations. We hope that our response has convinced you of the merits of our submission.
>
> Please do not hesitate to contact us if there are other clarifications or experiments we can offer. Thanks a lot!
>
> Best,\
> Authors

---

### Official Review · Reviewer_ZHhB · 2022-07-11

**Rating:** 6
**Confidence:** 4
**Soundness:** 3 good
**Presentation:** 3 good
**Contribution:** 3 good

**Summary:**

This paper incorporates the video-language modeling into continual learning. To overcome the challenges in continual learning, this paper proposes a novel cross-modal MoCo-based model with bidirectional momentum update (BMU).

**Questions:**

Could the author conducted more experiments on the multi-task transfer learning part?

**Limitations:**

yes

**Strengths And Weaknesses:**

strengths:
1) This paper provides the a new continual video-language modeling setting.
2) The extensive experiments demonstrate the effectiveness of the proposed method.

weaknesses:
1) This paper focuses on the video-language modeling in continual learning, however, the experiments only conducted on the text-to-video retrieval task. Since the proposed framework is two-stream architecture and is suitable for adjusting to the other video-language tasks, the experiments could be further investigated into multi-task transfer learning[1] in VLM domain, instead of on the single task.
2) The setting of two momentum encoder branches(local and global) is too complicated and the gains from global branch in table1 seems not significant.

[1] A deep hierarchical approach to lifelong learning in Minecraft, AAAI17

---

> ### Author Response · Authors · 2022-08-01
> **Response to Reviewer ZHhB**
>
> Thank you for the positive comments and insightful suggestions.
>
> **Q1. This paper focuses on the video-language modeling in continual learning, however, the experiments only conducted on the text-to-video retrieval task. Since the proposed framework is two-stream architecture and is suitable for adjusting to the other video-language tasks, the experiments could be further investigated into multi-task transfer learning (A deep hierarchical approach to lifelong learning in Minecraft, AAAI17) in VLM domain, instead of on the single task. Could the author conducted more experiments on the multi-task transfer learning part?** \
> **A1.** Good advice! As the reviewer mentioned, we focus on continual video-language modeling (CVLM) in this paper. Concretely, we choose to build our CVLM benchmark on the fundamental task (i.e., video-text retrieval), since video-language pre-training with streaming data is a realistic problem of practical significance. Meanwhile, the five video-text datasets of our benchmark come from different domains, which makes the CVLM setting even harder. Therefore, our proposed CVLM setting is both important and challenging. That said, it is still a good advice for us to investigate multi-task transfer learning using our BMU-MoCo. We then train a baseline model DrQ [a] and its BMU version DrQ+BMU under a multi-task transfer learning setting with three different visual control tasks of DMcontrol (which can also be called "continual reinforcement learning"). We obtain the evaluation rewards of the final models on all three tasks (i.e., Walker-Stand, Walker-Walk, and Walker-Run) as follows:
>
> | Method | Walker-Stand | Walker-Walk | Walker-Run |
> |----|  -----:  | -----:  | -----: |
> |DrQ [a]|  540 | 595 | 383 |
> |DrQ+BMU| $\bf679$ | $\bf837$ | $\bf413$ |
>
> We can observe that our BMU strategy can effectively alleviate the catastrophic forgetting problem in multi-task transfer learning (i.e., continual reinforcement learning). These results demonstrate the general applicability of our BMU strategy.
>
> **Q2. The setting of two momentum encoder branches(local and global) is too complicated and the gains from global branch in Table 1 seem not significant.** \
> **A2.** Thanks for pointing this out. We can see from Table 1 that BMU-MoCo (local) is indeed effective since it beats all the competitors. On top of BMU-MoCo (local), the global momentum encoders are used to further verify the effectiveness of BMU (i.e., updating encoders with momentum encoders to review past knowledge). This is supported by the fact that BMU-MoCo (local+global) performs better than BMU-MoCo (local). We believe that the gains from global branch over BMU-MoCo (local) are really remarkable since the results obtained by BMU-MoCo (local) are already good enough.
>
> [a] Yarats, D., Kostrikov, I., \& Fergus, R. Image augmentation is all you need: Regularizing deep reinforcement learning from pixels. ICLR 2020.

---

### Official Review · Reviewer_6RoA · 2022-07-18

**Rating:** 4
**Confidence:** 3
**Soundness:** 2 fair
**Presentation:** 2 fair
**Contribution:** 2 fair

**Summary:**

The authors propose a new continual video-language modeling (CVLM) setting, where models are supposed to be sequentially trained on five widely-used video-text datasets.
The authors propose the BMU-MoCo, a cross-modal MoCo-based model with a novel bidirectional momentum update (BMU) strategy.
To further boost our BMU-MoCo, a pair of global momentum encoders are maintained by the same BMU strategy.
The proposed method is tested on the proposed benchmark, and shows promising results.

**Questions:**

1. Equation (16) between Line 185 and 186: how is this contrastive loss implemented? Typically, such cross-entropy loss is implemented with log-softmax function to stablize the computation. However, in Equation (16), it seems not straightforward to use log-softmax to compute this loss. If computed by softmax and then log, is it stable?

2. LIne 290, "even outperforms ER-ring using 10% memory (about 200GB under our CVLM setting)". I'm curious that how this baseline that requires 200GB memory was implemented. Could the authors provide more details on the implementations of all these baselines. From my perspective, it would be a better contribution if the authors can make the CVLM benchmark and its baselines solid, compared with the newly proposed method.

**Limitations:**

Yes.

**Strengths And Weaknesses:**

Strength:
On the newly promised benchmark CVLM, the authors implemented several previous well-known baselines. This is a good effort.

Weakness:
The proposed method is only tested on the proposed benchmark. The authors may want to first make the proposed benchmark solid, or to show effectiveness of the proposed method on other continual learning benchmarks.

---

> ### Author Response · Authors · 2022-08-01
> **Response to Reviewer 6RoA**
>
> Thank you for the constructive comments and suggestions.
>
> **Q1. The proposed method is only tested on the proposed benchmark. The authors may want to first make the proposed benchmark solid, or to show effectiveness of the proposed method on other continual learning benchmarks.** \
> **A1.** Thanks for pointing this out. To our best knowledge, we have for the first time studied the continual video-language modeling (CVLM) setting and built a benchmark for CVLM which contains five video-text datasets (with 500K video-text pairs in total). In fact, the workload of constructing such benchmark is extremely heavy, since we have to re-implement 6 recent state-of-the-art continual learning competitors for CVLM along with our newly proposed BMU-MoCo and train all of them on five video-text datasets. Concretely, the total training time on such benchmark (with 6 baseline methods and 2 proposed methods) is around 7 days with 8 Tesla V100 GPUs (i.e., 2 months with 1 Tesla V100 GPU). In addition, as the reviewer suggested, we also apply the proposed BMU-MoCo to other continual learning settings such as unsupervised continual learning (UCL) to further verify its effectiveness. Concretely, we give the results of UCL for image classification (under the same setting as in LUMP [a]):
>
> | Method   |  Cifar-10 | Cifar-100 | Tiny-ImageNet |
> | -------- | ---------:| ---------:| -------------:|
> | Finetune |     90.11 |     75.42 |         71.07 |
> | DER [b]  |     91.22 |     77.27 |         71.90 |
> | LUMP [a] |     91.00 |     82.30 |         76.66 |
> | BMU-MoCo | **92.80** | **83.81** |     **77.45** |
>
> We can observe that our BMU-MoCo is still effective under the UCL setting and outperforms all competitors (their results are directly copied from LUMP [a]). We have additionally provided the results of UCL in Table 4 of the supplementary material.
>
> **Q2. Equation (16) between Line 185 and 186: how is this contrastive loss implemented? Typically, such cross-entropy loss is implemented with log-softmax function to stabilize the computation. However, in Equation (16), it seems not straightforward to use log-softmax to compute this loss. If computed by softmax and then log, is it stable?** \
> **A2.** Equation (16) is actually the log-softmax loss implemented as "softmax and then log" in Pytorch, which is the cross-modal version of the original contrastive loss in the well-known MoCo [c]. To show that this contrastive loss is indeed stable, we plot the training process (i.e., the change of contrastive loss) in Figure 2 of the supplementary material.
>
> **Q3. Line 290, "even outperforms ER-ring using 10\% memory (about 200GB under our CVLM setting)". I'm curious that how this baseline that requires 200GB memory was implemented. Could the authors provide more details on the implementations of all these baselines. From my perspective, it would be a better contribution if the authors can make the CVLM benchmark and its baselines solid, compared with the newly proposed method.** \
> **A3.** Sorry for the confusion. As stated in Lines 227--230 of our main paper, all rehearsal-based competitors (including ER-ring) have 10K stored video-text pairs as their memory buffer. This buffer requires 200 GB to save these extra data pairs. Moreover, as the reviewer suggested, we have added more implementation details of all these baselines in Section 4 of the supplementary material.
>
> We wish that our response has addressed your concerns, and turns your assessment to the positive side. If you have any questions, please feel free to let us know during the discussion stage. We appreciate your suggestions and comments! Thank you very much!
>
> [a] Madaan, D., Yoon, J., Li, Y., Liu, Y., \& Hwang, S. J. Representational continuity for unsupervised continual learning. ICLR 2022.\
> [b] Buzzega, P., Boschini, M., Porrello, A., Abati, D., \& Calderara, S. Dark experience for general continual learning: a strong, simple baseline. NeurIPS 2020.\
> [c] He, K., Fan, H., Wu, Y., Xie, S., and Girshick, R. Momentum contrast for unsupervised visual representation learning. CVPR 2020.

---

> > ### Comment · Reviewer_6RoA · 2022-08-07
> > **thank authors for the reply**
> >
> > 1. It's good to see the extra results on UCL. I'm afraid that it may need another peer review to assess the soundness and significance of the new results.
> > 2. The authors may not understand my question. The log-softmax implementation requires only 1 exponential term on the numerator (the number above the line in a common fraction), but in Equation (16) there are two exponential terms. More details of the implementation with the log-softmax function will be useful.
> > 3. "This buffer requires 200 GB to save these extra data pairs." In this case, this buffer can not saved in memory. How does the author implemented the ER-ring with such large buffer?

---

> > > ### Author Response · Authors · 2022-08-07
> > > **Response to post-rebuttal feedback**
> > >
> > > Thanks for your post-rebuttal feedback! For the new questions, our answers are given as follows:
> > >
> > > **Q1. It's good to see the extra results on UCL. I'm afraid that it may need another peer review to assess the soundness and significance of the new results.** \
> > > **A1.** Thanks. Our extra results on UCL are convincing, since we have strictly followed the UCL setting proposed in LUMP (Representational continuity for unsupervised continual learning, ICLR 2022). More importantly, we can clearly observe the performance gain over the state-of-the-arts obtained by our BMU-MoCo, demonstrating the effectiveness of our BMU-MoCo in other continual learning settings. Overall, we think that our new results are convincing and important for the continual learning community.
> > >
> > > **Q2. The authors may not understand my question. The log-softmax implementation requires only 1 exponential term on the numerator (the number above the line in a common fraction), but in Equation (16) there are two exponential terms. More details of the implementation with the log-softmax function will be useful.** \
> > > **A2.** Good suggestion. In PyTorch, we regard the two terms on the numerator in Equation (16) as one term, and then define our contrastive loss exactly the same as the standard log-softmax function. Note that such contrastive loss is known as the multiple-positives infoNCE loss (our loss is two-positives infoNCE loss), which has been widely used in recent works on contrastive learning, such as [1].
> > >
> > > **Q3. "This buffer requires 200 GB to save these extra data pairs." In this case, this buffer can not saved in memory. How does the author implemented the ER-ring with such large buffer?** \
> > > **A3.** Sorry for the confusion. This buffer can be directly saved in memory, since our machine has 500 GB memory in total.
> > >
> > > [1] Miech, A., Alayrac, J. B., Smaira, L., Laptev, I., Sivic, J., \& Zisserman, A. End-to-end learning of visual representations from uncurated instructional videos. CVPR 2020.

---

> > > > ### Comment · Reviewer_6RoA · 2022-08-07
> > > > **more details**
> > > >
> > > > 2. "multiple-positives infoNCE loss (our loss is two-positives infoNCE loss)". The paper you attach is not open accessible... I guess that your Equation (16) and other equations involving with two-positives may be not correct. You may refer to your code to check whether it exactly matches with Equation (16).
> > > > 3. By "200G memory", do you mean RAM or disk space?

---

> > > > > ### Author Response · Authors · 2022-08-09
> > > > > **Thanks a lot for your insightful discussion and looking forward to your post-rebuttal rating!**
> > > > >
> > > > > Dear Reviewer 6RoA,
> > > > >
> > > > > Thanks for your feedback again, which will help us further improve our final paper!
> > > > >
> > > > > **Q2. "Multiple-positives infoNCE loss (our loss is two-positives infoNCE loss)". The paper you attach is not open accessible... I guess that your Equation (16) and other equations involving with two-positives may be not correct. You may refer to your code to check whether it exactly matches with Equation (16).** \
> > > > > **A2.**
> > > > > Sorry for the inaccessible literature and we have replaced it with an accessible one. We agree that training stability is vital and there only requires one exponential term on the numerator in the original InfoNCE loss. Nevertheless, in our BMU, we utilize global encoders to further preserve earlier knowledge in addition to local encoders and there exist two positive samples for each video/text to obtain our loss $L_{final}$. Therefore, we slightly modified the InfoNCE loss with multiple positive targets following [1, 2] (two terms on the numerator), which is consistent with the MIL-NCE [1] loss. We have also double-checked the code and made sure it was consistent with Equation (16).
> > > > >
> > > > >
> > > > > **Q3. By "200G memory", do you mean RAM or disk space?** \
> > > > > **A3.**
> > > > > Yes, 200G memory means the RAM. Since memory-based methods need to be trained with memory data (which is supposed to be loaded into the RAM), these methods bring a huge computation resources cost. Though our machine has 500 GB RAM which can easily handle it, these methods still make it harder to apply in real-world applications. Note that our BMU only needs 0.5 GB extra RAM in total to save the global momentum encoders. Therefore, it is much more computational friendly.
> > > > >
> > > > > *Thanks for your time again! Please don’t hesitate to let us know, if there are any additional clarifications we can offer. Look forward to your post-rebuttal rating!*
> > > > >
> > > > > Best, \
> > > > > Authors
> > > > >
> > > > > [1] Miech, A., Alayrac, J. B., Smaira, L., Laptev, I., Sivic, J., \& Zisserman, A. End-to-end learning of visual representations from uncurated instructional videos. CVPR 2020.\
> > > > > [2] Huo, Y., Ding, M., Lu, H., Fei, N., Lu, Z., Wen, J. R., \& Luo, P. Compressed video contrastive learning. NeurIPS 2021.

---

> ### Author Response · Authors · 2022-08-07
> **Looking forward to your post-rebuttal feedback**
>
> Dear Reviewer 6RoA,
>
> Thanks again for your insightful suggestions and comments. As the deadline for discussion is approaching, we are happy to provide any additional clarifications that you may need.
>
> In our previous response, we have carefully studied your comments and made detailed responses summarized below:
>
> * Clarified the proposed benchmark for the continual video-language modeling (CVLM) setting.
> * Conducted additional experiments on unsupervised continual learning (UCL) to show the effectiveness of the proposed method on other continual learning benchmarks.
> * Added visualization results to show the stability of our contrastive loss implemented.
> * Provided more details on the implementations of all the baselines.
>
> We hope that the provided new experiments and additional explanations have convinced you of the merits of our submission.
>
> Please do not hesitate to contact us if there are other clarifications or experiments we can offer. Thanks a lot!
>
> Best,\
> Authors

---

### Author Response · Authors · 2022-08-09
**General Response: Contributions and New Experiments**

We sincerely appreciate all reviewers’ time and efforts in reviewing our paper. We are glad to find that reviewers generally recognized our contributions:

* **Model.** Proposing a novel cross-modal MoCo-based model with bidirectional momentum update for continual learning [6RoA, ZHhB].
* **Setting.**  Introducing a new setting of the Video-Language Modeling task in a continual learning scenario [6RoA, ZHhB, oj9S].
* **Experiment.** The experimental results on the proposed benchmark are promising [6RoA, ZHhB, oj9S].
* **Writing.** The Paper is well-structured [oj9S]; the contribution is clearly stated and supported in the experimental section. [oj9S];

And we also thank all reviewers for their insightful and constructive suggestions, which help a lot in further improving our paper. In addition to the pointwise responses below, we summarize supporting experiments added in the rebuttal according to reviewers’ suggestions.

**New Experiments**

* Comparison among different methods on unsupervised continual learning (UCL) setting [6RoA].
* More results by applying our BMU-MoCo for continual reinforcement learning [ZHhB].
* The training stability analysis of the final loss [6RoA].

We are glad to see that our response address most of Reviewer oj9S’s concerns. We hope that our pointwise responses below could clarify all reviewers’ confusion and alleviate all of their concerns. We'd like to thank all reviewers’ time again.

---

### Meta-Review · Area_Chair_Liw3 · 2022-08-31

**Recommendation:** Accept
**Confidence:** Less certain

**Metareview:**

This work presents a study on continual video-language modeling. In addition to the modeling side of things (BMU-MoCO), the authors construct a new benchmark in which they compare a number of existing methods. While I think it's great that the authors came up with a new benchmark, it's always a somewhat difficult analysis when a paper comes up with both a new benchmark and a method that beats the previous methods on this new benchmark. This is a shared concern with at least one of the reviewers. I do note that computational limitations make it difficult for the authors to thoroughly test on many other benchmarks. The authors do provide some UCL results in the rebuttal, which strengthen their case.

All in all, I have to agree with reviewer ZHhB that the method is somewhat complicated and that the gains from the global branch seem overstated. I found the methods part hard to follow as well. All in all, the work does seem interesting and important, but perhaps I am more convinced about the benchmark rather than the method. As is, I would still recommend it for acceptance, but I feel it would need a good amount of work to improve clarity of the exposition and ideally more robust empirical evidence (that doesn't involve benchmarks created by the authors themselves).



**Award:**

No

---

### Decision · Program_Chairs · 2022-09-14

Accept